# Medical Device Advances in the Treatment of Glioblastoma

**DOI:** 10.3390/cancers14215341

**Published:** 2022-10-29

**Authors:** Cher Ying Foo, Nimrah Munir, Ashwin Kumaria, Qasim Akhtar, Christopher J. Bullock, Ashwin Narayanan, Richard Z. Fu

**Affiliations:** 1Imperial College School of Medicine, Imperial College London, Fulham Palace Rd., London W6 8RF, UK; 2QV Bioelectronics Ltd., 1F70 Mereside, Alderley Park, Nether Alderley, Cheshire SK10 4TG, UK; 3Department of Neurosurgery, Queen’s Medical Centre, Nottingham University Hospitals, Nottingham NG7 2UH, UK; 4School of Medical Sciences, Faculty of Biology, Medicine and Health, University of Manchester, Michael, Smith Building, Dover St., Manchester M13 9PT, UK; 5Department of Neurosurgery, Manchester Centre for Clinical Neurosciences, Salford Care Organisation, Northern Care Alliance NHS Foundation Trust, Salford Royal, Stott Lane, Salford M6 8HD, UK

**Keywords:** Glioblastoma, GBM, medical device, convection enhanced delivery (CED), blood-brain barrier (BBB), sonodynamic, ultrasound, magnetic hyperthermia, laser interstitial therapy (LIT), brachytherapy, photodynamic therapy (PDT), electric field therapy (EFT)

## Abstract

**Simple Summary:**

Novel therapeutic modalities using medical devices have the potential to deliver effective treatment against the most aggressive form of primary brain tumour, Glioblastoma, for which there is currently no cure. These devices not only offer methods to enhance the effect of currently available therapies but also aim to provide innovative ways to deliver better therapeutic outcomes for this deadly disease. This review highlights recent innovations in this promising, growing field, and is intended to serve as a primer for clinicians, engineers and scientists interested in translating such technologies for the improvement of patient outcomes.

**Abstract:**

Despite decades of research and the growing emergence of new treatment modalities, Glioblastoma (GBM) frustratingly remains an incurable brain cancer with largely stagnant 5-year survival outcomes of around 5%. Historically, a significant challenge has been the effective delivery of anti-cancer treatment. This review aims to summarize key innovations in the field of medical devices, developed either to improve the delivery of existing treatments, for example that of chemo-radiotherapy, or provide novel treatments using devices, such as sonodynamic therapy, thermotherapy and electric field therapy. It will highlight current as well as emerging device technologies, non-invasive versus invasive approaches, and by doing so provide a detailed summary of evidence from clinical studies and trials undertaken to date. Potential limitations and current challenges are discussed whilst also highlighting the exciting potential of this developing field. It is hoped that this review will serve as a useful primer for clinicians, scientists, and engineers in the field, united by a shared goal to translate medical device innovations to help improve treatment outcomes for patients with this devastating disease.

## 1. Introduction

Glioblastoma (GBM) is the most common and aggressive primary malignant brain tumour, accounting for more than 60% of all brain tumours in adults [1]. The global incidence of GBM is approximately 3.21 per 100,000 population and has been increasing over the last decade [2,3]. Despite some therapeutic advances over the years, GBM patients continue to have a dismally poor prognosis with a 5-year survival rate of around 5% and a median survival of approximately 15 months [4,5,6].

The aggressiveness of GBM is characterized by its extensive tumour infiltration, microvascular proliferation, and high genomic instability. Histological characteristics of GBM include marked hypercellularity, microvascular proliferation and necrosis with pseudo-palisading features [7]. The hypoxic microenvironment of GBM also upregulates the expression of genes that facilitate angiogenesis, which enhances the proliferation and adaptation capabilities of GBM [8,9]. In addition, GBM contains self-renewing, tumourigenic cancer stem cells (CSCs), which are primarily responsible for therapeutic resistance and represent a therapeutic target [10,11,12]. The inter- and intra-tumour heterogeneity driven by these CSCs and by the tumour microenvironment collectively contribute to its resistance against the standard radio- and chemotherapy, greatly limiting the efficacy of current therapeutic options [13,14]. 

The current standard of care for GBM includes surgery, followed by radiotherapy and chemotherapy [15]. There is evidence to support the principle of maximal safe resection which seeks to achieve maximal cytoreduction during surgery, with view to increasing survival [16]. Unfortunately, complete tumour resection is usually impossible due to its highly invasive nature and the need to preserve eloquent brain tissue. The remaining tumour cells invariably infiltrate the normal brain region contiguous to the tumour, leading to progression or recurrence [17]. In addition, damage to the neighboring normal tissues with administration of radiotherapy is unavoidable due to its nonspecific cytotoxicity [18]. This is coupled with the additional challenges posed by the blood-brain barrier (BBB), which inhibits the permeation of drugs from the bloodstream into the brain, greatly reducing the delivery of chemotherapeutic agents [19]. Despite the multimodal therapeutic strategies, the prognosis of GBM remains dismally poor, with about 70% of GBM patients experiencing disease progression within one year, and a five-year survival rate of less than 5% [20]. Therefore, development of new therapeutic strategies against GBM is paramount to improve the outcomes of this devastating disease. Emerging pharmacotherapies for GBM have been reviewed previously [21,22,23,24,25]. However, given that clinical outcomes for GBM lag so far behind many other cancers, over the last few decades researchers have turned to technologies and methodologies from engineering and the physical sciences, beyond the more usual pharmaceutical preserve of the cancer field, in the search for effective new treatments. These technologies have resulted in a number of pioneering medical devices for the treatment of GBM (Figure 1). 

This review aims to provide a comprehensive overview of the clinical applications of current and emerging medical devices that directly deliver or augment therapy for GBM. Notably, we recognize that there have been recent advances in the development of intra-operative technologies to improve resection outcomes, such as the use of fluorescence guided resection [26]. and handheld Raman spectroscopy [27]. However, because these are surgical adjuncts rather than being therapeutic medical devices themselves, we have considered them beyond the scope of this current review. 

Broadly, therapeutic medical devices can be classed into two categories (1) Medical devices used to enhance current therapeutic modalities and (2) Medical devices that deliver novel therapeutic modality against tumour cells (Figure 2). We shall focus on clinical studies that have been published over the last two decades, highlighting both the promise and challenges of these treatment modalities and providing an outlook on potential future developments.

## 2. Medical Devices Used to Enhance Current Therapeutic Modalities

Medical devices that enhance current therapeutic strategies mainly work by disrupting/circumventing the BBB, such as convection-enhanced delivery (CED) and ultrasound-mediated BBB opening (BBBO) aim to deliver higher therapeutic drug concentrations in the brain and minimizing off-target adverse toxicity associated with systemic drug delivery (Figure 3) [28]. 

The BBB is a protective barrier composed of a continuous layer of endothelial cells connected by tight junctions that separates the systemic circulation from central nervous system (CNS) tissues [29]. It plays a crucial role in regulation of CNS homeostasis and is the first line of defense against toxins and microorganisms [30]. However, it is also a significant obstacle to the delivery of therapeutics to the brain as only lipid soluble molecules smaller than 400–500 Da can naturally cross the BBB, which excludes 98% of drugs [31]. Therefore, the lack of effective drug delivery and sufficient bioavailability within brain lesions has contributed to the high mortality of brain tumours relative to other cancers.

### 2.1. Convection-Enhanced Delivery 

Delivery of therapeutic agents into the brain to treat GBM has been an ongoing challenge for many years due to the presence of the BBB, which greatly limits the entry of therapeutics into the brain parenchyma [32]. CED is a treatment modality that allows targeted, local infusions of drugs directly into the tumour bed, bypassing the BBB. First conceptualized by Bobb et al., in 1994 for local delivery of macromolecules in the CNS [33]. It works by surgically placing catheters into the tumour and distributing the infusate directly into the tumour bulk by establishing a positive pressure gradient using extracranial infusion pumps thereby establishing a convective transport (Figure 3A) [33,34]. This method produces a high local drug concentration while limiting systemic toxicities and could potentially supersede the conventional diffusion-based approaches which rely solely on the compound’s concentration gradient and diffusivity and often only be able to achieve a limited volume of distribution of a few millimetres [35,36]. 

Since its inception, CED has been explored clinically with the use of a variety of therapeutic candidates and here we review noteworthy clinical trials using different agents including chemotherapeutic agents, viral vectors, and monoclonal antibodies (Table 1).

### 2.2. CED with Chemotherapeutic Agents

In 2020 Wang et al., reported the first clinical trial that demonstrate the feasibility and safety of CED of carboplatin in 10 patients with recurrent high-grade gliomas [38]. The authors reported that the median progression-free survival (PFS) and overall survival (OS) were 2.1 and 9.7 months, respectively, which compares favorably to previous trials using systemically administered carboplatin, in which median OS was around 6 months [44,45,46]. Patients reported an improvement in terms of seizure frequency and severity post-treatment. 

Likewise in the same year Upadhyayula et al., reported a long term follow up from a single institutional experience of a Phase 1B clinical trial investigating the use of Topotecan (TPT) CED in 10 GBM and 6 anaplastic astrocytoma patients [39,47]. They found that 11 out of 16 patients demonstrated either early response or pseudo-progression and these patients also had a significantly improved OS. Two patients became long-term survivors with a survival from the time of treatment of over 10 years. Moreover, over 75% of patients reported improved neurocognitive functions in terms of processing speed and memory, as well as improved quality of life at 8 and 16 weeks post-treatment.

### 2.3. CED with Viral Vectors

Oncolytic viruses have emerged as a novel therapeutic approach to GBM treatment, which utilizes native or genetically modified viruses that selectively replicate within tumour cells [48]. In 2014, Kicielinski et al., reported the first clinical trial of CED of a virus in patients with brain tumours [40]. They conducted a multicentre phase 1 trial in 15 patients with recurrent malignant glioma using the oncolytic reovirus, which was dosed at 1 × 108 to 1 × 1010 tissue culture infectious dose 50 (TCID50). No dose-limiting toxicity (DLTs) or maximum tolerated dose (MTD) was reached, demonstrating that the intratumoural infusion of reovirus is both safe and well tolerated. Furthermore, no grade 3 or 4 adverse events were reported, further supporting its safety profile. The median survival was 140 days (range 97–989) and median time to progression (TTP) was 61 days (range 29–150 days). The authors also found that the use of CED resulted in higher volume of distribution of reovirus compared with simple inoculation alone, as evident from the increased volume of T2 hyperintensity, which supports an increased distribution of reovirus over simple inoculation. However, the authors did not comment if this difference between the two administration approaches was statistically significant. 

In a recent study, van Putten et al., demonstrated the safety of Delta24-RGD (DNX-2401) administered by CED in a phase 1 trial on 20 patients with recurrent GBM [42]. Delta-24-RGD is oncolytic adenovirus with a 24 base pair deletion in the viral E1A genomic region, which renders the adenovirus unable to replicate in normal cells, but capable of replicating in cells with disrupted Rb pathway, which is present in more than 90% of gliomas [49]. This makes it an excellent candidate for specific targeting of glioma cells and its oncolytic potency has been previously demonstrated in several preclinical studies [50,51,52]. Furthermore, local Delta24-RGD treatment was also found to induce T-cell-mediated antitumour responses and established a protective immune memory [53]. The median PFS was 82 days (range 29–287 days), with a median OS of 129 days (range 68 days to more than 7 years). Study-related serious adverse events (SAEs) were mostly related to the increased intracranial pressure caused by inflammation-related edema or viral meningitis; however these symptoms were all transient. 

### 2.4. CED with Monoclonal Antibody

Antibody therapies are also being explored for CED applications such as OS2966, which is a humanized and de-immunized monoclonal antibody that targets the CD29/β1integrin, which is highly expressed in GBM and implicated in several hallmarks of cancer including growth, proliferation, invasion, angiogenesis, and drug resistance [54,55]. OS2966 has demonstrated excellent preclinical efficacy in GBM animal models and has been granted orphan drug designation by the FDA for the treatment of GBM. 

Nwagwu et al., have proposed the first in-human clinical trial investigating the use of CED to deliver OS2966 to treat high-grade gliomas [43]. Prior to each infusion, a gadolinium contrast agent will be added to enable real-time MRI visualization to confirm delivery of infusate to the targeted site. The trial is currently underway and is estimated to be completed by the end of 2023 (NCT04608812).

Overall, clinical studies have demonstrated the feasibility and favorable safety profile of CED. However, there are various challenges that need to be addressed before CED can fully realize its therapeutic potential. From a technical perspective, it is important to determine the type and design of catheter to optimize delivery, reduce infusate reflux, and achieve accurate catheter placement [56]. From a clinical perspective, challenges include the selection of the patient groups who are mostly likely to benefit and tolerate the treatment, management of neurological complications during and post-treatment and lastly, with currently only one FDA-approved chronic implantable subcutaneous pump based CED (Synchromed II, Medtronic; Minneapolis, MN, USA) in early phase 1B trial (NCT 03154996), the need for chronic infusions to maintain therapeutic drug concentrations and sustainable long-term benefits are major challenges still to be overcome [43,46,55]. 

### 2.5. Ultrasound-Mediated BBB Opening

One of the most promising technologies in recent years for BBBO is focused ultrasound (FUS) (Figure 3B). FUS devices have generated considerable interest within the scientific community for their potential to disrupt BBB. Currently, there are three different ultrasound devices available on the market—ExAblate 4000 (InSightec, Haifa, Israel), NaviFUS (NaviFUS Corp., Taipei, Taiwan), and SonoCloud (CarThera, Paris, France). Multiple preclinical studies have demonstrated the feasibility of FUS-mediated BBB modulation in optimising drug delivery for the treatment of brain tumours. This technique has been used to facilitate BBBO and subsequent delivery of a wide range of chemotherapeutic agents such as doxorubicin, carboplatin, and temozolomide (TMZ), resulting in delayed tumour growth and improved survival times [57,58,59,60]. Encouraged by the significant success in preclinical studies, several clinical trials of FUS-BBBO with various devices have been conducted in recent years to explore their efficacy and safety in humans (Table 2). 

A non-invasive approach for BBBO is the ExAblate system, which is an extracorporeal fixed stereotactic frame-based MRI-guided device used to deliver FUS to the CNS. This transcranial system couples a hemispheric transducer array to MRI, enabling intraoperative imaging and real-time acoustic feedback determining the sonication parameters [66]. 

The first clinical trial investigating BBBO using the ExAblate system in patients with malignant brain tumours was reported by Mainprize et al., in 2019 [61]. They demonstrated the feasibility and safety of transient BBB opening in the tumour tissue followed by systemically administered doxorubicin and TMZ, with an immediate 15–50% increase in contrast enhancement on T1-weighted MRI resolution. Furthermore, tissue liquid-chromatography mass spectrometry analysis also demonstrated greater concentration of liposomal doxorubicin and oral TMZ in brain regions where BBB disruption (BBBD) occurred compared to areas without BBBD, further validating the feasibility of improved delivery by FUS BBBD. Most importantly, the procedure was well-tolerated, with no adverse clinical or radiologic events related to the procedure, and all patients were discharged on the same day of their procedure. 

Park et al., investigated the therapeutic effect of BBBD with the ExAblate system on GBM patients during standard adjuvant TMZ chemotherapy [62]. They reported a 1-year survival rate of 100%, and no recurrence in four out of the six patients for 15 months. Furthermore, none of the patients had immediate or delayed BBBD-related complications during the 15-month follow-up period. Although two of the six patients had a recurrence at 11 and 16 months of follow-up, their PFS was 100% and 83.3% at 6 months and 1 year respectively. This is approximately twice the average PFS for GBM patients at 6 months (66.5%) and 1 year (46.8%), as reported by a recent meta-analysis [67].

One practical limitation with the ExAblate system may be to do with the longevity of effect of FUS BBBD; whilst varying FUS parameters can generate a greater effect of BBBD, a study by Samiotaki and Konofagu found the maximum length of BBBD effect of their tested FUS parameters to be 72 hrs, with a mean of 8 hrs [68]. Whilst GBM patients undergoing chemotherapy will frequently receive daily administrations, the same cannot be said of MRI, where the usual monitoring regime would involve scans monthly. The ExAblate system is essentially powered by MRI and requires the use of an MRI suite. To ensure continuous BBBD throughout a course of chemotherapy with the ExAblate system (which logically would lead to the greatest clinical efficacy), patients would have to undergo FUS in an MRI suite at least every 3 days. This would greatly increase the per patient use of MRI time, leading to possible congestion of such facilities and increased costs to hospital systems through radiographer time amongst other impacts.

SonoCloud is an implantable ultrasound device that can be inserted within the skull bone during surgical resection and can be repeatedly activated using a transcutaneous needle connection system prior to chemotherapy administration. The first phase 1/2a clinical trial using the SonoCloud technology was performed by Carpentier et al., in patients with recurrent GBM treated with intravenous carboplatin [63]. The authors found that the BBB was disrupted at acoustic pressure levels up to 1.1 MPa without detectable adverse effects on radiologic (MRI) or clinical examination, indicating that this technology is well tolerated in GBM patients. 

In a subsequent study, Idbaih et al., reported an increase in both median PFS and OS of 4.11 and 12.94 months, respectively, in GBM patients with BBBD, which is markedly higher than 2.73 months and 8.64 months in patients without BBBD [64]. This is consistent with previous studies which reported a PFS of 2–3 months and OS of 6–9 months in GBM patients treated by carboplatin alone [69,70]. The HR for PFS with clear BBB disruption after at least one sonication group was 0.39 (*p* = 0.03), and 0.49 (*p* = 0.09) for survival. Therefore, this suggests a potentially greater efficacy of carboplatin when used in combination with ultrasound-induced disruption of the BBB. 

NaviFUS is a novel frameless device that involves the use of a neuronavigational system to steer in real-time the transcranial ultrasound energy precisely and repeatedly toward targeted CNS regions [71]. NaviFUS has several innovative features, including a short procedure time of less than 15 min, a mobile device that does not require an expensive intraoperative MRI suite, and does not require skull fixation, which obviates the potential discomforts and delays associated with a stereotactic frame. Furthermore, its incorporation with the neuronavigation system maintains accuracy within an error of deviation of less than 3 mm [72]. 

In 2021, Chen et al., reported the first-in-human, prospective, open-label, phase 1 clinical trial investigating the feasibility and safety of NaviFUS treatment in patients with recurrent GBM [65]. The authors found a statistically significant signal intensity change (SIC) in BBB permeability immediately and 24 h after NaviFUS treatment on the contrast-enhanced T1 MRI (CE-T1). BBB permeability in all patients was showed to return near baseline within 24 h post-treatment. They also reported no adverse events related to FUS treatment; thus, the treatment was determined to be safe and tolerable for all patients in this study. 

The effect of FUS technology on BBBO and its efficacy and safety upon chronic application is still in its early stages. This needs to be studied further in future clinical trials, with greater attention to optimisation of various parameters related to volume of BBBO achieved which could play a major role in diffuse tumours such as GBM, and the target drugs that are going to be used [73]. 

## 3. Medical Devices Used to Deliver Novel Anti-Tumour Therapeutic Modalities

Medical devices that deliver anti-tumour therapeutic modality can be classed into two categories based on the need for surgical intervention 1) Non-invasive medical devices—such as Sonodynamic therapies (SDT), Magnetic hyperthermia-mediated cancer therapy (MHCT), high-frequency focused ultrasound (HIFU) and Electric field therapy (EFT) and 2) Invasive medical devices- such as Carmustine wafers, brachytherapy, Laser interstitial thermal therapy (LITT), Photodynamic therapy (PDT) and implantable EFT.

### 3.1. Non-Invasive Medical Devices 

The following medical devices do not require surgical intervention to provide their treatment (Figure 4).

#### 3.1.1. Sonodynamic Therapy 

Sonodynamic therapy (SDT) is an emerging modality for non-invasive treatment, which combines ultrasound with sonosensitizers to produce a localized cytotoxic effect via generation of reactive oxygen species [74] (Figure 4A). Sonosensitizers are chemical compounds that selectively accumulate in tumour cells, and their therapeutic activity are activated by ultrasound irradiation [74]. This allows for precise targeting of tumour cells, with negligible damage to surrounding healthy brain cells. Several sonosensitizers have been investigated in pre-clinical studies, two of these, PPIX and fluorescein, are most widely used due to their superior safety profile and selective accumulation in tumour cells (Figure 4A) [75,76]. 

Aminolevulinic acid hydrochloride (5-ALA) is a precursor of protoporphyrin (PPIX), which selectively accumulates in and sensitizes malignant glioma cells. The compound itself is non-toxic but can induce apoptosis of cancer cells when activated by ultrasound waves [77]. To date, several in vitro and in vivo studies using SDT combined with 5-ALA induced PPIX as a sonosensitizer in gliomas have been reported. Early preclinical studies demonstrated the effectiveness of SDT with 5-ALA induced PPIX and FUS irradiation for in vivo treatment of deep-seated intracranial C6 glioma in rats [78,79]. Selective tumour destruction and tumour growth inhibition were obtained by the non-thermal effect of FUS, which was enhanced by 5-ALA induced PPIX, without causing damage to surrounding normal brain tissue. Furthermore, the reduction in tumour volume was significantly higher in the SDT group compared to the control groups. In a subsequent study, Bilmin et al., reported significant 5-ALA induced PPIX -mediated SDT cytotoxic effects on rat RG2 glioma cells in vitro [80]. 

Other than 5-ALA induced PPIX, fluorescein (FL) is another sonosensitizer that is used for SDT. It is a biosafe xanthene dye with fluorescent properties, selective accumulation in brain lesions, and rapid washing out from vessels and healthy tissues [75,81]. These features make it a suitable compound for guiding surgical resection of malignant gliomas and more recently, an excellent candidate for SDT. Similar to 5-ALA induced PPIX, upon irradiation by ultrasound waves, FL is transferred to an excited state and generates ROS, thus exerting its anti-tumour activity [82]. In a recent study, Prada et al., demonstrate for the first time the cytotoxic effects of SDT with FL in C6 rat glioma cells [83]. They reported a high degree of FL accumulation within the tumour areas, with a nearly threefold increase in intratumoural epifluorescence signal over background. In addition, SDT with FL significantly inhibited C6 glioma growth across all three FUS exposure conditions of different acoustic intensities. 

A significant step forward has been made in 2021 by Raspagliesi et al., who reported the first intracranial MRI-guided SDT with FL and 5-ALA induced PPIX in a large animal model using the ExAblate system [84]. The porcine model allows for a more precise target definition, in a more similar way to the technique that would be used for clinical purposes in humans, compared to the more commonly used small rodent models. The authors demonstrate the feasibility and safety of SDT from a clinical, radiological, and histopathological point of view with both 5-ALA induced PPIX and FL as sonosensitizers for in vivo application of SDT. 

While clinical evidence of SDT in treatment of gliomas remains limited, two clinical trials are currently underway to investigate the use of intravenous 5-ALA (NCT 04559685) and oral 5-ALA (NCT 04845919) in combination with FUS in GBM patients [85]. 

#### 3.1.2. Magnetic Hyperthermia Therapy

Magnetic hyperthermia therapy (MHT) builds upon the principles of localized HT where temperature in a local region of the body is elevated above baseline by involving injection of magnetic nanoparticles (MNP) into the tumour and subsequent application of an alternating magnetic field (AMF), resulting in an increased intratumoural temperature and thermal ablation of tumour cells (Figure 4B). 

MagForce Nanotechnologies (Berlin, Germany) has developed a MHT system, NanoTherm, which consists of an aqueous deposition of superparamagnetic iron oxide nanoparticles (SPIONs) with an iron concentration of 112 mg/mL. These biocompatible nanoparticles are injected directly into the tumour site and are subsequently exposed to an alternating magnetic field to generate heat for ablation of tumour cells. The treatment area of the AMF applicator (NanoActivator) has a diameter of 20 cm, a magnetic field strength of up to 18 kA/m, and a field frequency of 100 kHz [86]. NanoTherm therapy has received approval by the European Medicines Agency (EMA) as a medical device for the treatment of brain tumours since 2011 [87]. 

Several clinical trials have demonstrated the efficacy and safety of intratumoural thermotherapy using aminosilane-coated SPIONs (NanoTherm) in GBM patients. In a phase 1 study, Maier-Hauff et al., demonstrated the feasibility of thermotherapy using MNP in 14 patients with recurrent GBM [88]. SPIONs were injected directly into the tumour site and exposed to an alternating magnetic to induce particle heating. They demonstrated that MNPs could generate the therapeutic intratumoural temperature (42.4–49.5 °C) and was well tolerated in patients with no neurological complications. 

In a subsequent phase 2 study, the authors investigated the combination of MNP and external bean radiotherapy in 59 recurrent glioblastoma (rGBM) patients [89]. The AMF parameters used were identical to those in the phase 1 study, and the median peak intratumoural temperature was 51.2 °C. They reported a significantly prolonged OS following diagnosis of tumour recurrence of 13.4 months, which is greater than twice that of patients treated with chemotherapy alone (6.2 months) [15]. 

More recently, Grauer et al., investigated the tolerability and efficacy of the intracavitary thermotherapy using SPIONs combined with radiotherapy in 6 rGBM patients [90]. The authors created a technique to coat the walls of the resection cavity with 2–3 layers of NanoTherm using a hydroxycellulose mesh and fibrin glue to increase the stability of the nanoparticle film and create sufficiently high SPION concentrations. Thermotherapy consisted of six weekly sessions, which lasted 1 h each. They reported a median overall OS and PFS of 8.15 and 6.25 months respectively. Earlier interventions were also associated with more favourable outcomes. Patients treated at first recurrence had longer survival times than those treated at second recurrence or later (23.9 months vs. 7.1 months). There were no adverse effects during active treatment, however after 2–5 months, all patients experienced significantly increased perifocal edema around the MNP deposits, leading to clinical deterioration. This was temporarily managed with dexamethasone treatment, but four of the patients required neurosurgical interventions to remove the nanoparticle deposits together with the adjacent granulation tissue, after which their condition improved.

MagForce has reported that one course of NanoTherm costs roughly €23,000 per patient. However, there are no studies in the literature examining its cost-effectiveness due to a lack of efficacy data, making direct comparisons of NanoTherm to other medical device treatments impossible [91]. 

Moving forward, it is imperative to optimize MHT parameters, including ensuring accurate intratumoural heating and precise temperature control at the tumour site as well as minimize perifocal oedema. One practical consideration in the use of MNP may be their potential to generate interference in MRI imaging; if the particles persist within region of the tumour resection cavity, MRI artefacts generated by the very large difference in magnetic susceptibility between tissue and the MNPs would likely obscure large volumes of the surrounding tissue (including the tumour resection margins) on imaging. If MHT were to be used to treat newly diagnosed GBM, clinicians may be concerned that such interference could limit their ability to detect and monitor tumour recurrence and thereby impact decision making in ongoing patient care. In the event of a truly efficacious MHT therapy, such a practical consideration is likely to be forgiven. However, if survival outcomes for MHT are only equivalent to other emerging treatment modalities, it may factor into the decision making of clinicians over which therapy to pursue. 

#### 3.1.3. High-Intensity Focused Ultrasound 

High-intensity focused ultrasound (HIFU) is a therapy where under MRI guidance a stereotactic device is used to distribute high intensity energy (100–10,000 W/cm^2^) through the skull, which can increase tissue temperature up to approximately 65 °C, thereby inducing precise thermal ablation at target tumour sites with minimal effect on surrounding tissues [92,93]. This increase in temperature causes coagulative necrosis and protein denaturation within a few seconds, producing immediate and localized tumour cell death (Figure 4C) [94]. 

In 2006 Ram et al., reported a Phase 1 clinical study involving 3 patients with recurrent GBM who underwent MRI-guided high-intensity focused ultrasound (MRgHIFU) thermal ablation [95]. They showed that primary lesions responded to the MRgHIFU with immediate changes in the contrast-enhanced T1-, T2-, and diffusion-weighted MRI scans, in addition to thermocoagulation on histological examination. However, one patient had an adverse outcome caused by thermal ablation of brain parenchyma outside the target in the pathway of transmission of the ultrasound waves, leading to neurological deficits. 

In 2010, McDannold et al., reported on a Phase 1 clinical trial in 3 GBM patients, which demonstrated for the first time that ultrasound beams can be focused noninvasively through the intact skull and heating was visualized using real time MR temperature imaging (ExAblate 3000 system) [96]. The patients were treated at acoustic power levels of 800 W and 650 W, and focal heating in the targeted tumour site was induced to an overall maximum temperature of 51 °C for 20 s sonication time. Unfortunately, the study was limited by the low power of the FUS device (650–800 W), which was insufficient in reaching the ablation focal thermal threshold of 55 °C and therefore was unable to achieve complete tumour ablation.

In a more recent study, Coluccia et al., reported brain tumour ablation with transcranial MR-guided FUS in a 63-year-old patient with centrally located recurrent GBM [97]. Thermal tissue ablation was achieved by transmitting pulses of FUS of 10–25 s duration with acoustic power of 150–950 W into the target tumour site. A total of 25 sonications were applied and intraoperative MR thermometry identified 17 of the 25 sonication’s as capable of coagulation, with temperature peaks in the range of 55 °C–65 °C. Immediate post-interventional diffusion-weighted MRI images revealed multiple bright lesions representing the thermally coagulated tissue in the targeted tumour volume. No new treatment-related neurological deficits were observed, and the patient also showed improvement in neurological symptoms during the follow-up period. 

Overall, HIFU appears to be a promising technology for brain tumour ablation. However, as the clinical experience of HIFU is currently limited to small case series, further clinical trials are required to evaluate the safety and efficacy of HIFU for thermal ablation of brain tumours. Currently, two trials (NCT01473485 and NCT00147056) are underway on 10 patients each, looking at the safety and feasibility of MRgHIFU thermal ablation using the ExAblate transcranial system with results expected at the end of 2022 which hopefully will provide further insight regarding the feasibility of HIFU. Potential drawbacks of the ExAblate system on patient through-flow in MRI suites discussed previously would also apply here.

#### 3.1.4. Electric Field Therapy 

EFT, variously also known as Tumour Treatment Fields (TTF) or Intratumoural Modulation Therapy (IMT), is a recent treatment modality that has shown efficacy both in preclinical studies and clinical trials of GBM patients (Table 3) [98,99,100,101,102,103]. The use of low intensity, intermediate frequency and alternating electric fields, has demonstrated extended PFS and OS in patients with primary and recurrent GBM (Figure 4D) [98,104].

EFT has received widespread attention due to its remarkable ability to specifically influence the fate of proliferating cells, whilst leaving non proliferating cells unaffected. Although the exact mechanism by which EFT exerts anti-tumour effects is not fully understood, an anti-mitotic effect has been proposed whereby they are thought to alter the tumour cell polarity, ultimately disrupting normal polarization and depolarization of mitotic spindle microtubules, causing dielectrophoretic dislocation of intracellular macromolecules and organelles during cytokinesis. This leads to miotic and cell membrane disruption and ultimately to apoptosis [99,105,108]. It has also been shown to target the DNA damage repair and breast cancer 1–mediated (BRCA1-mediated) pathways promoting replication stress and thereby being more susceptible to radiation therapy [109,110]. A recent study by Chen et al., demonstrated a role for EFT in activating anti-tumour immunity both preclinically in syngeneic murine GBM models and clinically in patients treated with EFT by promoting the production of immune-stimulating proinflammatory and interferon type 1 cytokines in tumour cells [111]. Interestingly, glutamatergic neurotransmission has recently been shown to contribute to GBM progression, and EFT may selectively disrupt such tumour initiated signaling to explain its efficacy in part [112,113]. 

To date, a single EFT device, Optune (Novocure Inc., Jersey City, NJ, USA), has received regulatory approval for the treatment of supratentorial GBM in adults aged 22 or over [98,104]. Initial concerns over the use of EFT in patients under the age of 22 may be due to the possibility of its anti-mitotic effect impacting the development of the juvenile brain. Nevertheless, EFT use in pediatric GBM has recently been reported in individual case reports and small case series, and small trials assessing safety and efficacy are currently recruiting [114,115].

Optune consists of a portable electric field generator, carried in a backpack or satchel, connected via cables to non-invasive insulated transducer arrays affixed to the scalp. Alternating electrical fields at frequency of 200kHz are passed sequentially between paired sets of these transducer arrays on opposite sides of the head to generate an electrical field intensity of at least 1V/cm throughout the tumour to the supratentorial brain regions with a recommended treatment duration of 18 h per day. Parameters including intensity of the electric field and frequency are pre-set and are software controlled [105]. 

Clinical trials have demonstrated the safety and efficacy of Optune in the treatment of GBM. The first pilot trial (EF-07) began in 2004 using the Novo-TTF-100a system (forerunner to Optune) in 10 patients with recurrent GBM and the effectiveness of the treatment was compared to recurrent GBM historical controls. This was the inaugural instance of EFT being used as a strategy to treat GBM [105]. These findings led to the launch of a randomized phase 3 clinical trial (EF-11) to investigate the efficacy of EFT as a monotherapy in patients (*n* = 120) with recurrent GBM tumours compared to physicians’ choice chemotherapy alone (*n* = 117). Findings revealed no significant difference in OS or PFS in patients receiving EFT vs. chemotherapy. Therefore, it was concluded that using EFT was just as effective as using chemotherapy in recurrent GBM treatment, but with fewer side effects and overall improvement of quality of life [104]. Based on these findings, the first instance of FDA approval for the use of EFT as a treatment for recurrent GBM following standard of care chemotherapy was provided. A real-world analysis of its use in recurrent GBM patients over a two-year period (2011–2013) further confirmed its safety and tolerability [116]. 

Subsequently, a multicenter randomized phase 3 clinical trial (EF-14) assessed the efficacy of Optune plus adjuvant TMZ (*n* = 466) vs. TMZ alone (*n* = 229) in newly diagnosed GBM (ndGBM) patients [98]. All patients completed initial surgical intervention and radio-chemotherapy prior to randomization and received standard maintenance TMZ with or without EFT. The combination of EFT and TMZ was demonstrably more effective compared to TMZ alone with noted PFS of 6.7 months vs. 4 months (*p* < 0.001) and OS was 20.9 months vs. 16.0 months (*p* < 0.001). The 5-year survival rate for patients receiving EFT plus TMZ was 13% vs. 5% for TMZ alone (*p* < 0.001). One of the major limitations of the conclusion of the EF-14 trial may be that only patients that were progression free at the completion of chemoradiation were enrolled in the study, thereby removing the poor prognosis patients from the study [117]. Questions have also been raised regarding the understandable lack of a sham EFT group (as it would be unethical to ask patients to shave their heads and daily wear a non-functioning device) but this does also raise the possibility of a placebo effect biasing results [117]. An aspect of the completed clinical studies of Optune in ndGBM worth highlighting is the fact that patients completed standard of care TMZ and concomitant radiotherapy prior to commencing treatment with Optune. This represents a delay of approximately four to six weeks between surgery and commencing EFT—which may have limited its observed efficacy. Part of the reason for this delay may have been to allow for post-operative wound healing prior to the application of scalp transducer arrays. 

In the EF-14 trial, systemic adverse events were commonly related to chemotherapy use and were mostly absent in the EFT arm, suggesting the safety of electrical fields as a treatment modality [118]. The most common side effect noted in the EFT treated group was scalp irritation associated with the transducer arrays, where moderate irritation was observed in 52% of the patients and severe irritation in 2% [98]. However, a later study of 27 patients in Southern China reported dermatologic adverse events in as many as 82% of the patients [119]. One possible explanation for the increased incidence of dermatological issues could be the heat of the tropical climate in that region. Secondary analysis of trial data revealed that patients receiving EFT (alongside TMZ) presented improved survival and reported no negative influence on health-related quality of life (HRQoL) when compared to TMZ alone [120]. Nevertheless, with the primary determinant of HRQoL in a GBM patient being disease progression [121], one would expect an improvement in HRQoL in the EFT arm vs. TMZ alone considering the clinical results clearly demonstrate that EFT is effective in improving survival. In actuality, the data demonstrate no significant difference in HRQoL between the two arms. This could be the product of two opposing effects—the EFT treatment efficacy acting to improve HRQoL, whilst the impact of Optune treatment on patient lifestyle acting to decrease HRQoL to give no overall effect. This was observed in a study of a small cohort (*n* = 7) of patients undergoing Optune therapy, Olubajo et al., reported disruption of daily activities including showering, cooking, going out in the rain and sleep [122]. These lifestyle impacts resulted in one of the seven patients discontinuing treatment. Patients cited the weight & size of device, length of cables and inconsistent alarms (leading to sleep interruption) as being issues with the Optune device. Despite the reported issues, all patients in Olubajo’s cohort would recommend Optune to someone else, which may be reflective of the terrible prognosis faced by GBM patients, the survival benefit EFT has been shown to provide, and the lack of an alternative EFT device [122]. 

Following EF-14′s positive results, FDA approval was achieved for concomitant use of chemotherapy and EFT for ndGBM patients. Despite this, the adoption of Optune within the neuro-oncology community has been somewhat limited—in the first full year (2016) following its FDA approval for ndGBM, Optune received 2344 prescriptions in North America (predominantly the USA), by 2018 this had quickly grown to 3741 patients (~15% of new GBM cases in the USA). Optune has since received national reimbursement in Japan, Israel, Sweden and Germany [123]. However, in the years since 2018 there has been little further increase in the number of prescriptions with 3781 patients reported at the end of 2021 [124]. This plateau in Optune prescriptions could be due to multiple contributing factors including operational difficulties during the COVID-19 pandemic, low cost-effectiveness, limited understanding of the underlying molecular-level mechanism of action and the perceived limitations of its landmark randomized clinical trials [102,125]. Perhaps reflective of the design issues reported by Olubajo et al., we are aware of anecdotal evidence from clinicians and patient advocacy groups suggesting that some patients are unwilling to undergo treatment with Optune due to their concerns over its highly conspicuous design [122,126] and the need to carry the device throughout the day. Such concerns may also have an impact on the willingness of physicians to prescribe/recommend the therapy to their patients or the strength of that endorsement. Wick et al., also raises the point that, unlike almost any other therapy, the direct-to-patient model of Optune sees patients interacting with Novocure employees more regularly than with their clinical teams. Disruption of the established physician-patient relationship and concerns over the influence a commercial entity (with a conflict of interest) may have over patient decision-making regarding ongoing treatment may be another factor in the clinical willingness to recommend Optune [117]. 

Due to the entirely novel nature of EFT as a treatment modality, there are many outstanding questions regarding how its clinical application can be optimised. At the forefront of these is limited understanding of the possible synergistic effects between EFT and other treatment modalities, including Chemo-/Radio- therapy and some of the other technologies discussed in this review. An initial pilot trial demonstrated the safety and efficacy of the concurrent use of EFT, TMZ and radiotherapy in patients with ndGBM [106]. This has warranted the initiation of several phase 2 and 3 clinical trials to determine the safety and efficacy of this concurrent triple-modality therapy (EFT, chemotherapy, and radiotherapy) [106,107,127,128]. These trials are ongoing, and completion will provide valuable insight into its synergistic potential. 

Another area that requires investigation in order to optimise the efficacy of EFT is in dosing compliance; strong correlation between dose of EFT delivered and OS has been suggested by the EF-14 trial data, with patients receiving >22 h of daily EFT presenting a median OS of 28.7 months and 5 years survival rate of 29.3% [98]. This suggests that the therapeutic efficacy of EFT can be further improved by increasing daily treatment time, warranting further studies to investigate the potential benefit.

One hypothesis would be that all other factors being equal, the greatest survival benefit of EFT would be achieved through continuous 24-h a day treatment. However, the external design of Optune makes achieving 24-h therapy in a significant number of patients unlikely due to the need to daily affix electrodes, regular head shaving along with other considerations in patient lives such as bathing. One approach to address this issue would be through an implantable EFT device, which would theoretically enable continuous therapy whilst also bypassing the possible patient quality of life impacts of skin irritation and the conspicuous design of Optune.

### 3.2. Invasive Medical Devices 

The following medical devices require a minor or major surgical intervention to provide treatment and usually require the placement within the tumour bulk, or in the cavity left after surgical intervention (Figure 5).

#### 3.2.1. Carmustine Wafers

Intracranially implanted local chemotherapy using alkylating agent Carmustine (1,3-bis(2-chloroethyl)-1-nitrosurea (BCNU)), have been approved by FDA for recurrent high grade gliomas since 1996 and for new high grade gliomas since 2003 [129]. GLIADEL wafers are a biodegradable copolymer used to control the release of carmustine and are approximately 1.45 cm in diameter and 1 mm thick. Each wafer contains 7.7 mg of carmustine [1, 3-bis (2-chloroethyl)-1-nitrosourea, or BCNU] and 192.3 mg of a biodegradable polyanhydride copolymer. The recommended dose is to implant 8 wafers for a total of 61.6mg [130]. These wafers are applied to the resection margins with the aim of providing locoregional treatment, increased efficacy and reducing systemic toxicity (Figure 5A) [131]. 

Since its first use in the late 1990’s, various clinical trials have been carried out using Gliadel wafers for both ndGBMs and recurrent GBM. Westphal et al., in a phase 3 trial with BCNU wafers on high grade glioma (HGG) patients demonstrated an improved OS (13.9 vs. 11.6 months) with no change in PFS, when compared to placebo treated control [132]. 

A propensity matched French multicentre study by Pallud et al., in 2015, demonstrated that carmustine wafers implantation in combination with SOC demonstrated both an improved median PFS (12 vs. 10 months) and median OS (20.4 vs. 18 months) when compared to standard group [133]. 

Carmustine wafers have different reported costs in various markets. In the UK, carmustine wafers were assessed for recommendation by the National Institute for Health and Care Excellence (NICE) for use in the National Health Service (NHS). The assessment by NICE priced carmustine wafers at £6105 for 8 wafers, plus the additional cost of managing potential adverse effects whilst determining that the addition-al QALY (quality-adjusted life years) was equal to 0.107 (5.6 weeks). This placed the final ICER (incremental cost-effective ratio) value at £57,000 per QALY. NICE deemed carmustine wafers to lack evidence of cost-effectiveness, as the cost per QALY was far higher than its typical £30,000 per QALY threshold [134]. 

However, complications arising from these wafers have been a hindrance in their uptake and use within the clinical setting A study by Bregy et al., using data from 19 studies on 795 BCNU wafer patients reported a mean OS of 16.2 months and a staggering complication rate of 42.7%, prompting them to recommend not using the agent [135]. Furthermore, their high cost, reported high complication rates, and challenges of directly handling the agent by operating room staff have all contributed to reduced uptake of it in clinics.

#### 3.2.2. Brachytherapy

Currently, the mainstay of GBM therapy is a multimodal strategy of surgical resection followed by administration of adjuvant radiotherapy 4–8 weeks after the initial surgery [15,136]. Although external beam radiotherapy (EBRT) has been shown to be effective in improving local tumour control, it has several drawbacks, including a waiting period of 4–5 weeks for post-operative wound healing and recovery. During this time, 50–70% of GBM patients experience tumour regrowth adjacent to the resection cavity [137,138,139], which is associated with poor survival outcomes [140]. Notably, more than 80% of GBM recurrence occurs within 2 cm of the resection cavity, highlighting the importance of local control [141].

Brachytherapy, the implantation of interstitial radioactive isotopes in close proximity to the target tissue, has emerged as an attractive treatment option in this context, as it can be performed intraoperatively directly after tumour resection, allowing irradiation to begin immediately. In recent years, cesium-131 (131Cs) has emerged as a promising isotope for brachytherapy for GBM [142,143,144]. Table 4 outlines current clinical trials investigating the use of 131Cs for brain tumours. This isotope confers physical and dosimetric advantages compared with the previously used iodine-125. This is mostly attributed to its shorter half-life of 9.7 days compared to 59.4 days of I-125, therefore its rapid dose falloff minimizes radiation exposure to the surrounding normal brain tissue while maintaining therapeutic doses at the tumour site [145]. This has translated into more rapid dose delivery, improved efficacy, as well as a superior safety profile (Figure 5B) [64,69]. 

Several studies over the past few years have evaluated 131Cs brachytherapy tiles for the treatment of brain cancers [70,71,72]. The largest of these studies was a prospective trial by Wernicke et al., which reported 100% local freedom from progression (FFP) rate for all tumour sizes, as well as a regional 1-year regional FFP rate of 89% [148]. Furthermore, none of the enrolled patients developed radiation necrosis, therefore highlighting the efficacy and safety of 131Cs brachytherapy for patients. 

More recently, a permanently implantable device consisting of 131Cs radiation emitting seeds embedded within a resorbable collagen-based carrier tile (GammaTile, GT Medical Technologies, Tempe, AZ, USA) has received FDA clearance to treat brain tumours. As a form of surgically targeted radiation therapy (STaRT), GammaTile (GT) delivers a large dose of radiation to the tumour bed upon implantation, while sparing the surrounding tissue. Each tile measures 2 cm × 2 cm × 0.4 cm and contains four 131Cs radioactive seeds with a half-life of 9.7 days and photon energy of 30.4 KeV [149]. The low dose rate 131Cs seeds deliver 120–150 Gy at the matrix surface and 60–80 Gy at 5 mm depth, with rapid dose fall-off thereafter [145,146,150]. Its low dose delivery rate, combined with short half-life give rise to its favourable safety profile. The arrangement of seeds within the collagen carrier was designed to optimise delivery of the radiation dose. Compared to the traditional forms of brachytherapy which consists of directly implanted radioactive seeds, the collagen carrier secures the seeds equidistant from each other and minimises seed migration after implantation, allowing for uniform dose delivery. This circumvents the problem of non-uniform delivery of radiation and direct contact of the seeds with the brain parenchyma. More importantly, the tissue offset of 3 mm provided by the tile dimensions reduces the risk of focal necrosis around the sources, thereby minimising radiation-induced necrosis while maintaining local disease control [151]. 

Gessler et al., reported the first clinical series of the use of GT in treatment of recurrent GBM patients since its FDA clearance in late 2018 [146]. The study cohort consists of 22 recurrent GBM patients who underwent maximal surgical section followed by GT implantation. The median OS of the GT-treated cohort (24.4 months) was significantly higher than that of the control cohort (17 months). Similarly, the median progression-free survival (PFS) of 8.2 months was higher than that of patients without GT treatment (5.1 months). The use of GT has also achieved favourable local control of 86% and 81% at 6 and 12 months respectively. Importantly, none of the patients suffered from adverse radiation effects that required medical or surgical intervention. However, it is worth noting that this study has inherent limitations such as its small sample size and exclusion of GBM patients who recurred within six months of standard of care radiation therapy, thereby introducing bias in the survival prognostication. 

Another study by Budnick et al., involving seven patients with recurrent gliomas (2/7 with rGBM) reported that none of the enrolled patients have shown signs of radiation necrosis on surveillance imaging whilst receiving adequate radiation coverage [147], thereby demonstrating the potential of GT as a safe adjuvant for recurrent gliomas.

However, there are some practical limitations to GT. It has been found that tumour cells more than 5–8 mm distant to the resection cavity where GT is implanted are less likely to benefit from this therapy [145], which reduces its efficacy in the treatment of diffusely invasive cancers that extend beyond the radiation margins. It is likely that future advances in therapeutic efficacy against GBM will require a multifaceted treatment approach with GT and other forms of adjuvant therapy. Moving forward, larger prospective clinical trials are likely necessary to investigate the efficacy of GT compared with standard-of-care treatment and to identify patient groups who are likely to benefit the most from GT therapy.

#### 3.2.3. Laser Interstitial Thermal Therapy

Laser interstitial thermal therapy (LITT) is a novel thermal ablation treatment which uses a stereotactically guided laser-tip probe to deliver controlled thermal energy to tumour bulk. Real-time magnetic resonance imaging (MRI) thermometry is also used for continuous monitoring of the ablation zone [152]. This process occurs when photons emitted by the laser optical fibre are absorbed by tumour cell chromophores, resulting in chromophore excitation followed by release of thermal energy [153,154]. Once a sufficiently elevated temperature is reached, protein denaturation, cellular necrosis, and tissue coagulation occur (Figure 5C). As a minimally invasive procedure, LITT is a favorable alternative treatment option in patients with deep-seated or difficult-to-access lesions, or in those who could not tolerate an open surgical resection [154]. Table 5 outlines the current clinical trials evaluating the use of LITT for treatment of GBM. 

The first clinical trials of LITT on GBM patients were performed in 1994, which demonstrated a decrease of total lesion size (15–87%), highlighting the feasibility of LITT for the treatment of brain tumours [162]. Currently, there are 2 commercial LITT systems available: Visualase Thermal Therapy System (Medtronic, Fridley, MN, USA) approved by the FDA in 2009 after a phase I study in 4 patients with brain metastasis, and NeuroBlate System (Monteris Medical, Plymouth, MN, USA) which received FDA clearance in 2013 after a first-in-human study in 10 patients with unresectable recurrent glioblastomas [163,164]. The Visualase Thermal Therapy System uses a diffusing fiberoptic tip probe (980 nm at 15 W) combined with saline cooling, while the NeuroBlate system uses a diode laser in the Nd-YAG range (1064 nm at 12 W) [165,166].

Several studies have demonstrated the feasibility and safety of LITT in the treatment of patients with both ndGBM and rGBM. In 2013, the first-in-human Phase 1 study using NeuroBlate to assess the safety and efficacy of LITT in rGBM at 3 thermal dose levels was published [155]. The median survival time was 316 days (range 62–767 days), which compares favourably to the 90–150 day median survival typically observed for rGBM [167]. Similarly, the median PFS at 6 months is estimated to be greater than 30%, which is approximately twice that of rGBM. Although most patients were clinically stable and suitable for discharge within 48 h after the procedure, several complications were reported including intracerebral hemorrhage, deep venous thrombosis, pulmonary embolism, and neutropenia. 2 patients also developed serious neurological adverse events due to the procedure, both at the highest dose level, but were successfully managed with no long-term neurological deficit. Interestingly, both patients were noted to have unexpected patterns of thermal energy deposition although the maximum tolerated dose was not reached. This suggests the need for careful observation of the patterns of thermal deposition during LITT as well as additional intraoperative measures for such unexpected patterns.

In a subsequent study, Patel et al., reported their single-center experience of 102 patients with a variety of intracranial tumours, the largest series to date [156]. The authors reported that 14 patients (13.7%) developed new neurological deficits after the procedure, and of those patients, 64.3% (*n* = 9) had complete resolution of deficits and 7.1% (*n* = 1) had partial resolution of symptoms within 1 month, 14.3% (*n* = 2) had not had resolution of deficits at the most recent follow-up, and 14.3% (*n* = 2) died as a result of progression without resolution of deficits. For the 10 patients who had resolution of deficits, the authors found that their neurological symptoms were caused by the postoperative edema in the ablation bed, which resolved with steroids. In the 2 patients who died, 1 lesion was relatively large and the other was located within the midbrain and pons. Rapid development of malignant edema in both patients caused the global neurological decline. This shows that LITT, although minimally invasive, should be used with caution as thermal damage to critical and eloquent structures can lead to serious complications. In a recent study, Kamath et al., reported a median OS of 11.5 months and median PFS of 6.6 months following LITT treatment of 54 patients with GBM (41 of whom has recurrent disease) who were not candidates for open surgical resection [157]. They were able to achieve an average of 93% of tumour volume treatment within the yellow TDT line and 88% treatment with the blue TDT line, suggesting that LITT is an effective cytoreductive option. The overall median OS also compares favorably with the OS of chemotherapeutic regimens, which were reported to be around 9 months [168,169,170,171]. Therefore, the available evidence suggests that the utilization of LITT at GBM recurrence can potentially achieve a survival benefit. A large multicenter cohort study by Mohammadi et al., demonstrated the efficacy of LITT compared with biopsy alone for treatment of deep-seated ndGBM [158]. The authors reported that the overall OS of patients who underwent LITT was comparable to that of those who underwent biopsy alone (14.4 vs. 15.8 months). 

Beaumont et al., demonstrated that treatment with LITT in 15 patients with GBM of the corpus callosum achieved a PFS and survival post-LITT (SPL) of 3.4 and 7.2 months, respectively, and OS of 18.2 months [159]. This is comparable with the median survival achieved with maximal safe surgical resection (∼65% volume reduction) of such tumours [172]. In a subsequent retrospective cohort study of 6 patients, Shah et al., also highlighted the meaningful cytoreductive potential of LITT (mean PFS of 14.3 months) in patients with deep inaccessible gliomas who would otherwise be offered a stereotactic biopsy [173]. 

Many studies have suggested that LITT is best suited for tumours with diameters less than 3 cm, which corresponds to a tumour volume of approximately 14.1 cm^3^. Notably, a considerable number of post-LITT complications in large tumours have been reported, including malignant oedema, and increased intracranial pressure, necessitating urgent craniectomy and debulking of the tumour [160,174]. 

Nevertheless, these studies have altogether shown that LITT is a feasible and minimally invasive treatment for GBM and can be used as an alternative in patients in whom aggressive surgery is not an option. As LITT is becoming part of the neurosurgical armamentarium, it is important to consider the benefits and risks of this technology. The risks of LITT can be placed in 2 main categories: (1) laser insertion, in which haemorrhage is a known complication, and (2) thermal ablation, which could result in unintended thermal damage to surrounding structures, and is particularly problematic for ablation targets that lie close to critical neural structures [175]. We hope that future advances in thermal monitoring will result in refinement of techniques and improvement in clinical outcomes. 

There are also several benefits to LITT. Aside from the minimal invasiveness of the approach, real-time monitoring of both thermal distribution and damage allows for a high degree of therapy control [152]. Compared with conventional surgery, LITT is also associated with decreased morbidity, reduced postoperative hospital stay and overall cost [176]. Furthermore, as LITT has been found to increase the overall survival by 3.07 months at an additional cost of $7508 per patient, its ICER of $29,340/life years gained (LYG) is low for general international standards, falling below NICE thresholds for cost-effectiveness, whilst also much lower than the US thresholds which can vary from between $50,000—$150,000 per QALY. In general, LITT seems to demonstrate high levels of cost-effectiveness, which will ease adoption in different countries [176].

However, the current literature consists mainly of pilot trials, phase 1 trials and retrospective analysis. Larger prospective series are required to evaluate the efficacy of this technique. Efforts are also currently underway to standardize and optimize the LITT procedure as well as to identify patient populations and tumour characteristics (location and volume) that would benefit most from this minimally invasive approach. 

#### 3.2.4. Photodynamic Therapy

Photodynamic therapy (PDT) is a treatment modality that works through photoactivation of a light sensitive dye—photosensitizer (PS) that is incorporated into neoplastic cells. Once a photosensitizer accumulates within neoplastic cells, in the presence of oxygen, it can be excited by photo-irradiation using visible light of appropriate wavelength, which causes the PS to convert molecular oxygen into either a singlet state or triplet state [177,178,179]. PDT’s role in cellular death has been shown to be multipronged with it evoking the three main death pathways; apoptotic, necrotic and autophagic. It has also been shown to impact vasculature within and around tumour cells thus impacting tumour growth [179,180]. 

The use of PDT as a therapeutic option against cancer dates to late 1970s with report on the effect of Hematoporphyrin derivative (HPD) plus light on bladder carcinoma [181]. The first use of PDT against Glioma was in 1980 when Perria et al., used light (632.8 nm) to activate HPD in the post resection cavity of glioma patients [182]. Over the following 4 decades, various other PDT trials have taken place specifically on GBM patients. One of the major criteria for PDT to work is the selectivity and penetrance of PS used, and to this end various PS have been tested over the years [183,184].

PDT can be divided into two types based on the surgical intervention needed—(1) **Interstitial PDT (iPDT)**—This is a minimally invasive procedure and performed on patients whose tumours are present in eloquent areas or on fragile patients who cannot undergo a craniotomy and (2) **Intracavitary PDT**—Brain tumours that can undergo complete or partial resection, PDT can be performed within the tumour cavity at the end of the surgical procedure (Figure 5D).

**Interstitial PDT**—The use of iPDT within glioma patients is usually performed via placement of multiple optical fibers with a diffuser within tumours using stereotactic coordinates [184]. Multiple technical considerations needed to be taken into account before iPDT, such as, the need to select the right geometry of light diffuser to attain maximum photobleaching of the target tumour, the number of diffusers required to achieve this, thorough planning on the placement of the said diffusers to negate any adverse effects on eloquent areas of the brain and finally the selection of the right tumour size and location to deliver effective and safe iPDT [183]. One of the major challenges has been achieving the balance between maximal photo stimulation of a given tumour volume and minimizing thermal injury to the normal brain tissue. To overcome this issue, modelling studies have determined that cylindrical diffusers had a better light distribution as well as reduced thermal sensitivity to normal tissue when compared to that of a flat diffuser [185]. However, a flat diffuser has an advantage when treating tumours close to eloquent areas of the brain as the light fluence delivered by it drops off more rapidly than when using a cylindrical diffuser [185].

To achieve optimal therapeutic effect of PDT, ‘advanced photobleaching’ of the PS is necessary. By definition, advanced photobleaching is achieved when ≥95% of PS undergoes photobleaching. To this end advanced dosimetric modelling of PS used should therefore be taken into consideration before and during surgical planning. 

A recent systematic review by Leroy H-A et al., on the use of iPDT against GBM over the last 3 decades demonstrates that overall the technique was safe and effective [179]. The study looked into 12 iPDT trials with over 250 patients receiving iPDT (68% of which were GBM) [184]. The trials mainly used either HPD or PPIX induced by 5-ALA as PS, with most studies after 2007 using the latter due to its better uptake and selectivity in tumour cells. The median wavelength used for the excitation of the PS in the 12 studies was 630 ± 5 nm and with a light intensity of 200 mW/cm diffuser length used, it was estimated that the potential therapeutic effect extended to 1 cm in diameter through the brain parenchyma [184]. In most cases 6 intracerebral fibers were used to deliver the therapy, with Beck et al., reporting that a distance of 9mm between the fibers was essential to reduce the possibility of a thermal effect between the fibers [186]. 

The median progression-free survival (PFS) was 14.5 months for de novo GBM and 14 months for recurrent GBM. The overall survival (OS) was 19 months for de novo GBM and 8 months for recurrent GBM. The discrepancy between PFS and OS for rGBM was due to the heterogeneity between the studies and not all studies reporting PFS and OS [184]. 

A recent single-center retrospective study on the use of iPDT as a salvage treatment for local recurrent malignant gliomas demonstrated promising outcomes with a time-to-treatment failure of 7.1 months and with a 2- and 5-year post-recurrence survival at 25% and 4.5%, respectively, [187]. Although highly promising the study has a few drawbacks, the biggest being that the study was based on a monocentric cohort of consecutive patients undergoing iPDT between 2006 and 2018, thereby not having a matched control group. A larger prospective study would be able to provide a better understanding on the therapeutic benefit of iPDT in rGBMs [187].

The systematic review by Leroy H-A et al., did point to specific criteria for prolonged survival (>2 years): (a) Preoperative Karnofsky PS > 70, (b) Complete response on early brain imaging (c) Well limited/spherical lesion (d) Tumour volume <5 cm^3^ and (e) Strong tumour PPIX uptake [184].

Of all these criteria, tumour volume is an extremely important factor in determining not only efficacy but also safety of iPDT. The median tumour volume from the 12 studies were found to be 12 cm^3^, with Kaneko et al., determining that a tumour volume of less than 5 cm^3^ was required to achieve complete response without persistent neurological deficits [188,189]. Larger tumours that undergo iPDT were associated with increased risk for neurological deficits and an increase in morbidity. This was observed in the Krishnamurthy et al., case series where the median tumour volume was 50 cm^3^, in which 41% (5 out of 12) of GBM patients developed permanent neurological deficit [190]. 

**Intracavitary PDT**—Intracavitary PDT or post resection PDT is performed after the maximal safe tumour resection has been undertaken in the operating room or in the post-operative recovery room. Cavitary PDT is usually performed by placing a balloon filled with diffusing liquid within the resection cavity with an external light source within it. The balloon is inflated to conform to the resection cavity thereby diffusing the light to all margins of the tumour resection cavity [183]. 

A pilot feasibility and safety study by Vermandel et al., (INDYGO trial) on 10 ndGBM patients was conducted in 2018 (NCT 03048240). Patients underwent maximal tumour resection guided by 5-ALA Fluorescence Guided Surgery (FGS) followed by intraoperative PDT. Postoperatively patients underwent adjuvant therapy (Stupp protocol). The interim analysis showed an actuarial 12-months progression-free survival (PFS) rate of 60% (median 17.1 months), and the actuarial 12-months OS rate was 80% (median 23.1 months). They also noted that a dose deposit of 200 J/cm^2^ at the balloon wall ensured the deposition of at least 25 J/cm^2^, which had been determined to be an optimal dose in preclinical studies [191]. No unacceptable or serious adverse effects were observed [191,192]. 

When applying PDT outside the operating room, a balloon diffuser is placed in the resection cavity and filled with a radio-opaque lipid emulsion until the cavity is filled. The PDT is then applied in the recovery room for a total of 5 treatments (one per day), after which the balloon diffuser is deflated and removed in the recovery room. 

Eljamel et al., conducted a single-center, randomized controlled phase III trial to evaluate porfimer sodium mediated PDT after 5-ALA FGS for ndGBMs. In the study, 13 patients underwent FGS followed by intracavitary placement of a balloon diffuser to provide repetitive PDT (1 session per day, 100 J/cm^2^ applied per session) for 5 days during the post-operative period. The control arm underwent FGS for tumour resection without PDT. The mean survival of patients in the PDT and surgery only groups was 52.8 weeks and 24.2 weeks, respectively (*p* < 0.001) [183,193]. 

Various complications related to PDT have been reported over the years. The majority of them relate to the PS and the systemic administration of it. Although this varies depending on the type of PS used, one of the most common risks related to all PS is that of retinal and cutaneous photosensitivity. This photosensitivity can last from several days to a few weeks depending on the PS used and during this time exposure to direct sunlight is harmful to the patient [194]. Apart from retinal and skin sensitivity, various other adverse events have also been reported including but not limited to post-operative hemorrhage, neurological deficit, deep venous thrombosis, infection, uncontrolled cerebral edema and even death [183]. 

Although various studies have pointed towards favorable outcomes when using PDT (both iPDT and intracavitary PDT), there remains a large dearth in concrete data on the length of the outcome. The lack of large RCT’s with a dedicated PS and dosing regimen as well as towards a standard control group makes interpretation difficult [183,184].

#### 3.2.5. Implantable Electric Field Therapy

External EFT in the form of Optune^TM^ as described above, has shown convincing survival benefit compared to SOC, through a series of sequential, large randomized controlled trials over the past decade in recurrent and ndGBM. Of the treatment modalities reviewed, it has demonstrated the biggest gain in survival metrics for GBM patients, with a 5-year survival of 13% compared to 3% in patients treated with SOC [195]. Despite this, adoption within healthcare systems remains limited as highlighted earlier.

An implanted approach may be able to overcome the limitations of the current device especially relating to issues of dosing compliance and potential quality of life concerns. It is also likely to provide significant advantages in terms of delivering the therapy directly to where it is needed to enable better local disease control (Figure 5E). Pre-clinical studies to date have demonstrated the potential and efficacy of implanted EFT in vitro and in vivo [100,101], and novel clinical approaches including device implantation in intravascular spaces using endovascular techniques or intraventricular/cerebrospinal fluid (CSF) spaces have been proposed [196].

Generally, modern medicine trends toward the use of minimally invasive therapy over the use of implantable medical devices admittedly due to the risks associated with surgical procedures. However, surgical resection very much remains a key component of the standard of care for GBM and delivering an implant during an existing cytoreductive surgical procedure may mitigate such concerns. There will of course be patients who will be unable to undergo surgical resection due to tumour location, age, or comorbidity, for which such a strategy may not be possible. In which case externally applied EFT therapy or approaches such as LITT remain viable alternatives following SOC treatment. 

Despite the promise of these studies and the potential benefits of an implantable EFT approach, considerable technical challenges remain to be solved to enable this form of therapy to be translated clinically. These include how to generate sufficient coverage of an effective electrical field through the tumour volume/surrounding brain, how to power an implantable system and how to implant a system safely surgically. Proposed solutions to some of these design challenges have included using ultrasound to power a stimulator [103] or using computational simulation techniques to plan the surgical placement of electrodes in a patient-specific manner [197,198]. Recent advances in advanced materials and biocompatible electrode designs may provide an opportunity to overcome these technical challenges, paving the way for a more efficient and precise use of EFT for the treatment of GBM via an implanted approach. 

## 4. Conclusions

Despite decades of research to improve patient outcomes, GBM remains a challenging disease to treat. Its molecular and cellular heterogeneity, critical location, and the BBB are critical constraints which limit the efficacy of current treatment options. In recent years, a range of novel device technologies have exhibited significant potential for the treatment of this aggressive cancer. While promising, many of these studies are limited by low enrolment numbers, limited follow-up, and potentially confounded by varying adjuvant therapies. Therefore, additional studies are necessary to investigate efficacy, establish optimal patient selection criteria, standardized surgical technique and treatment parameters of many of these technologies. More knowledge of the toxicity profiles, long-term stability, and safety data of these novel therapies are also necessary before widespread adoption by the clinical community is likely to occur.

For many of these devices, a greater understanding of the structure, physiology, and barrier properties of the BBB will allow optimisation of these device technologies in order to enhance the delivery of anticancer therapeutics. In conjunction, careful patient selection by clinicians will be key. For example, highly vascularized tumours might benefit more from FUS, as this technique requires systemic administration of microbubbles and drugs. Furthermore, FUS was found to suppress efflux transporters, which could potentially increase the accumulation of anticancer drugs within tumour tissue [199]. In contrast, CED is especially suited for tumours with a low vascular density, which minimise the excretion of the infused drugs into the vasculature, thereby allowing greater accumulation at the tumour site [200]. 

EFT has arguably shown the most promise to date of all the technologies highlighted in this review, with regard to increasing the OS of patients with this deadly disease. Nevertheless, ongoing studies to further understand the underlying mechanisms of EFT, particularly when used in conjunction with various emerging pharmacotherapies could help establish it as a key component in the standard of care patients receive [201]. Potential studies into variable sub-type vulnerability for example, could also pave the way for more personalized EFT approaches, potentially providing more effective treatment. Furthermore, moving to an implantable solution could overcome notable limitations of the external approach, including dosing compliance, patient lifestyle impact and the 4–6 week wait time before therapy commences, all of which may have a major impact on the efficacy of the therapy.

Finally, with the concurrent emergence of various medical device technologies as highlighted in this review, there may soon come a point in time where clinicians and healthcare payers will be faced with a difficult choice between various novel technologies, all of which have study outcomes reporting efficacy to varying degrees. It is therefore important going forward that a concerted and considered effort is made to undertake studies which attempt to identify what might be the best technology to use in combination with SOC as well as importantly, emerging molecular and immune-mediated therapies. Furthermore, stratifying future study patients in line with the recently updated 2021 WHO classifications which add further weight to the molecular underpinnings of disease such as isocitrate dehydrogenase (IDH) status, will be equally important [202]. This way, patients faced likewise with an increasing range of treatment options can be appropriately counselled, and their expectations managed. Thoughtful, collaborative engagement involving clinicians, patients, scientists, and the companies that develop these devices, ideally championed by respective neuro-oncology societies and patient charities, will help optimise the development of these promising approaches. Such will hopefully lead in the near future to meaningful treatment breakthroughs for patients with this devastating, incurable disease.

## Figures and Tables

**Figure 1 cancers-14-05341-f001:**
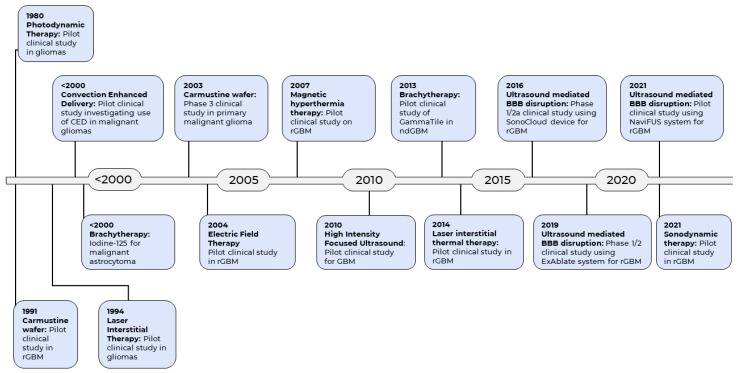
Historical timeline of the emergence of medical devices as therapeutic modality for GBM.

**Figure 2 cancers-14-05341-f002:**
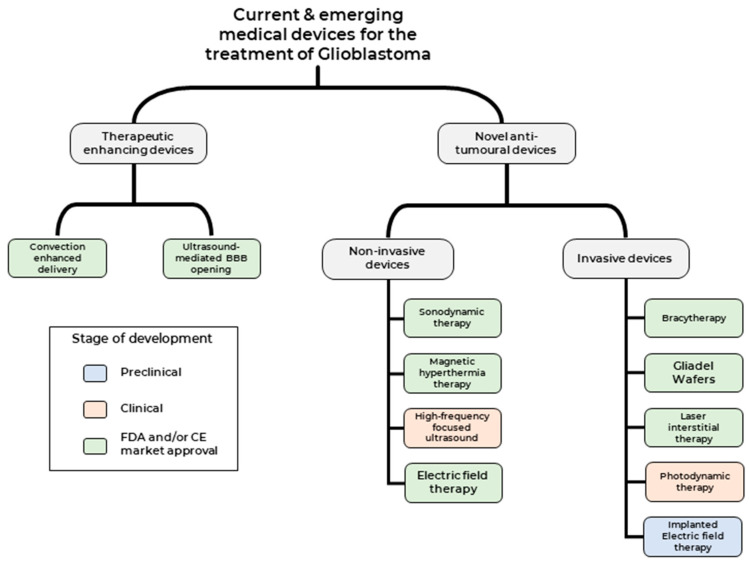
Current and emerging medical devices for the treatment of GBM.

**Figure 3 cancers-14-05341-f003:**
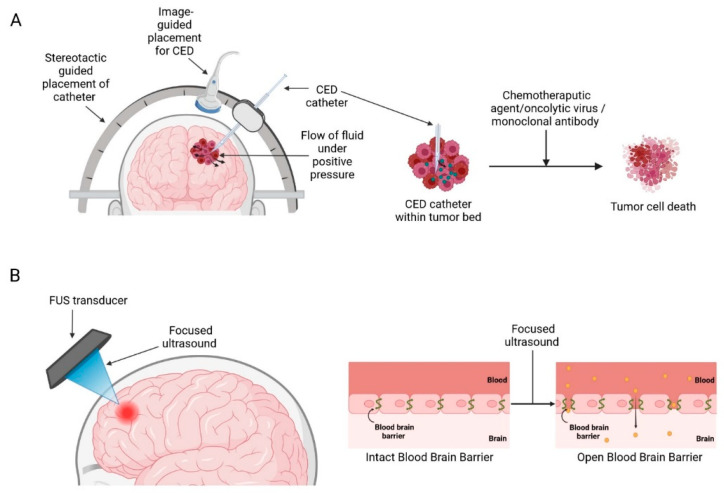
Medical devices used to circumvent/disrupt BBB: (**A**) Convection enhanced delivery (**B**) Focused ultrasound mediated blood-brain barrier opening. Created with BioRender.com accessed on 3 October 2022.

**Figure 4 cancers-14-05341-f004:**
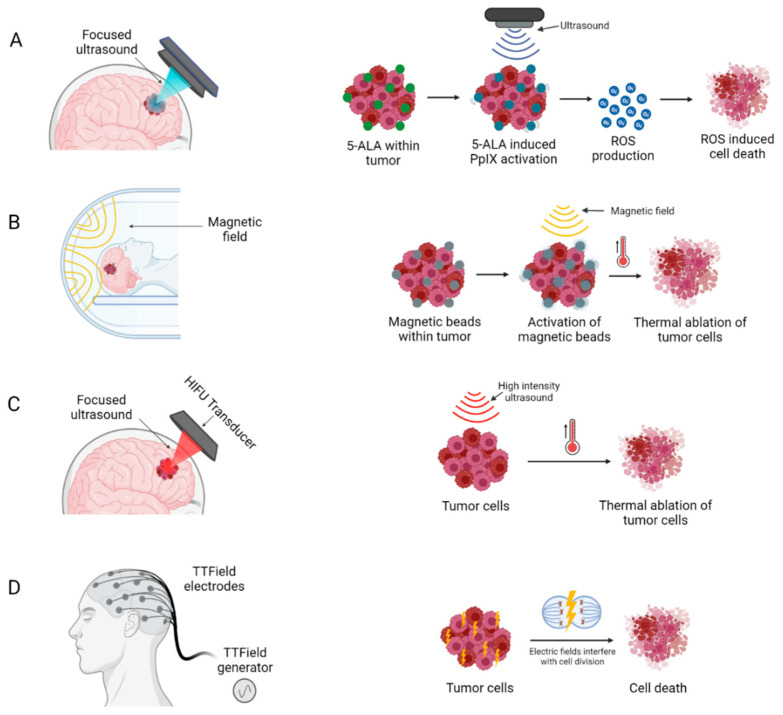
Medical devices delivering novel non-invasive anti-tumour therapeutic modalities: (**A**) Sonodynamic therapy (**B**) Magnetic Hyperthermia (**C**) High intensity focused ultrasound and (**D**) Electric field therapy. Created with BioRender.com accessed on 3 October 2022.

**Figure 5 cancers-14-05341-f005:**
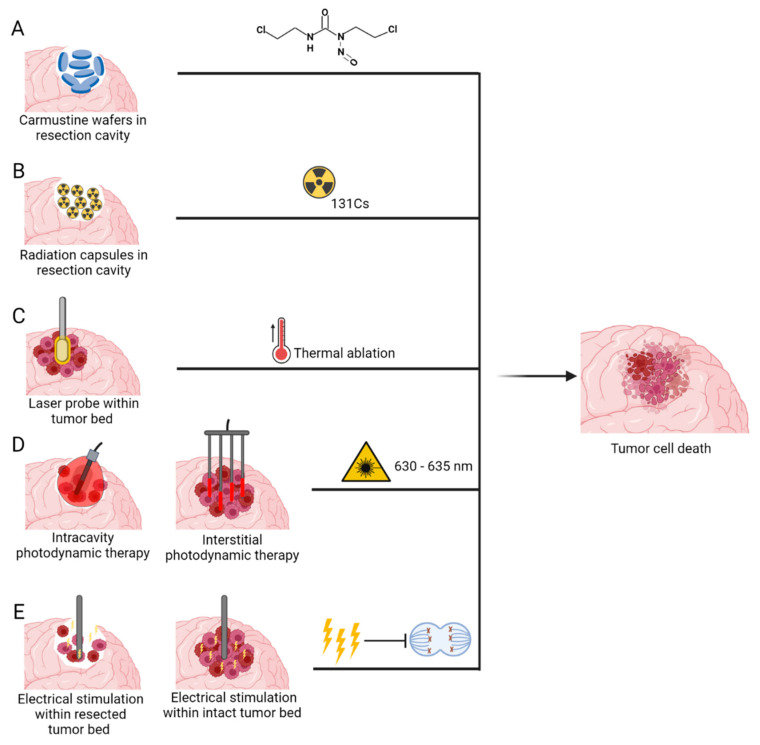
Medical devices delivering novel invasive anti-tumour therapeutic modalities: (**A**) Carmustine wafers (**B**) Brachytherapy (**C**) Laser Interstitial Thermal Therapy (**D**) Photodynamic therapy and (**E**) Implantable Electric Field Therapy. Created with BioRender.com accessed on 13 October 2022.

**Table 1 cancers-14-05341-t001:** Summary of clinical trials utilising CED to deliver therapeutic agents.

Study Design and Trial Ref	Indication	Therapeutic Agent	No. of Patients	Catheter	Infusion Rate (mL/h)	Volume Infused (mL)	Duration (h)	Median OS (Months)
**Chemotherapeutic Agent**								
Pilot trial NCT02278510 [37]	Recurrent HGG	Topotecan	3	2 Cleveland Multiport catheters	0.396	38	96	n/a
Pilot trial NCT01644955 [38]	Grade III/IV gliomas	Carboplatin	10	1–4 barium-impregnated CSF-ventricular catheters	0.75	54	72	9.6
Phase 1b dose-escalation study [39]	GBM	Topotecan	10	Implantable chronic infusion pump (Synchromed II, Medtronic)	0.2	n/a	100	n/a
**Oncolytic Virus**								
Phase 1 trial [40]	Grade III/IV gliomas	Purified reovirus	18	1–4 CED catheters (Phoenix Biomedical)	0.4	n/a	72	4.6
Phase 1 trial NCT01491893 [41]	Grade IV malignant glioma	PVSRIPO	61	Medfusion 3500 or 3010 infusion pump and catheter	0.5	3.25	6.5	12.5
Phase 1 trial [42]	Recurrent GBM	Delta24-RGD	20	2 CED catheters	0.2 to 0.3	n/a	44 to 66.7	4.24
**Monoclonal Antibody**								
Phase 1 trial NCT04608812 [43]	Grade III/IV gliomas	OS2966	n/a	Infuseon Cleveland Multiport Catheter	Max 0.3	Up to 4.8	Up to 4	n/a

**Table 2 cancers-14-05341-t002:** Summary of clinical trials evaluating the use of FUS devices to induce BBB opening.

Study Design and Trial Ref	Indication	Device	No. of Patients	Drug	FUS Parameters	Main Findings
Phase 1, single-arm trial NCT02343991 [61]	Grade III/IV gliomas	ExAblate Neuro 4000 (220 kHz)	5	Temozolomide or liposomal doxorubicin	4–15 W, 0.74% DC for 50s	Safe and effective opening of BBB, with immediate 15–50% increase in contrast enhancement, and resolution 20 h later
Prospective single arm trial NCT03712293 [62]	GBM	ExAblate Neuro 4000 (220 kHz)	6	Temozolomide	210s per target	Median survival has increased up to 14.6 months, the 2-year survival rate up to 27.2%, and the 5-year survival up to 10%
Phase 1/2a single arm trial NCT02253212 [63]	Recurrent GBM	SonoCloud (1.05 MHz)	17	Carboplatin	0.5–1.1 MPa	BBB was disrupted at acoustic pressure levels up to 1.1 MPa without detectable adverse effects
Single arm pilot trial NCT02253212 [64]	Recurrent GBM	SonoCloud (1.05 MHz)	21	Carboplatin	0.41–1.15 MPa	Patients with clear BBB disruption had an increased median OS of 12.94 months.
Phase 1, single-arm trial NCT03626896 [65]	Recurrent GBM	NaviFUS system (500 kHz)	6	n/a	Energy doses: 0.48, 0.58, 0.68 MI; total exposure time: 120s	Safe and reversible BBB opening using NaviFUS in patients with rGBM.

**Table 3 cancers-14-05341-t003:** Completed clinical trials for EFT devices.

Study Design and Trial Ref	Indication	Intervention	No. of Patients	Median OS (Months)	Median PFS (Months)	Main Findings
Phase 3 randomised trial (EF-14) NCT00916409 [98]	ndGBM	NovoTTF-100A device, chemotherapy	695 (466 TTFields, 229 chemo)	6.7 vs. 4.0	20.9 vs. 16.0	TTField improved OS with no further toxicity and negative effect on quality of life vs. chemo
Phase 3 randomised trial (EF-11) NCT00379470 [104]	rGBM	NovoTTF-100A device, chemotherapy	237 (120 TTFields, 117 chemo)	6.6 vs. 6.0	2.2 vs. 2.1	TTField just as effective as chemo but with reduced severe adverse reactions
Single arm pilot trial (EF-07) [105]	rGBM	NovoTTF-100A device, chemotherapy	10	14.3 (62.2 weeks)	4.9 (26.1 weeks)	Mild to moderate contact dermatitis. No device related serious adverse events
Single arm pilot trial, NCT03780569 [106]	ndGBM	NovoTTF-200A, Radiotherapy (60Gy), temozolomide	10	-	8.9	Grade 1–2 TTFields related skin toxicity.
Single arm Phase 2 trial, NCT01894061 [107]	rGBM	NovoTTF-100A device, Bevacizumab	23	10.5	4.1	

**Table 4 cancers-14-05341-t004:** Studies evaluating 131Cs brachytherapy in brain tumours.

Study Design and Trial Ref	Indication	No. of Patients	Local FFP (%)	Median OS (Months)	Complications (Total %)
Prospective case series NCT04427384 [146]	Recurrent GBM	22	81	24.4	CSF leak, DVT, seizure (13.6%)
[147]	Recurrent gliomas	7	n/a	n/a	Radiation necrosis (7.6%)

**Table 5 cancers-14-05341-t005:** Summary of studies evaluating the use of LITT for treatment of GBM.

Study Design and Trial Ref	LITT System	No. of Patients	Median Tumour Volume (cm^3^)	Median OS (Months)	Median PFS (Months)	Complications
[153]	NeuroBlate	34	10.13	Not reached	5.1	Neurological deficits (7), seizure (1), DVT (1), infection (2)
Phase 1 trial NCT00747253 [155]	NeuroBlate	10	6.8	10.4	n/a	Neurological deficits (2), ICH (1), DVT (2), PE (1) neutropenia (1),
Retrospective analysis [156]	Visualase	87	n/a	n/a	n/a	Neurological deficits (14), haemorrhage (3), refractory edema (5), infection (2), deaths (3)
Retrospective descriptive study [157]	NeuroBlate	54	12.5	11.5	6.6	cerebral oedema (3), seizures (3), hydrocephalus (1), infection (1), death (2)
Retrospective study [158]	NeuroBlate	28	9.3	14.4	4.3	Neurological deficits (6), ICH (2), DVT (1)
Retrospective analysis [159]	NeuroBlate	15	18.7	7.2	3.4	Neurological deficits (4), hydrocephalus (1), ventriculitis (1)
Pilot trial [160]	NeuroBlate	17	11.6	10.9	7.6	Transient aphasia (3), transient hemiparesis (3), DVT (1), meningitis (1)
Retrospective analysis [161] (Wright et al., 2016)	NeuroBlate	10	38	16.1	9.3	Neurological deficits (2), infection (3), hydrocephalus

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
