# Peer review of "Medical Device Advances in the Treatment of Glioblastoma"

_cancers, 2022, doi:10.3390/cancers14215341_

Round 1

Reviewer 1 Report

The authors provide an accessible, readable, and comprehensive review on the current state of research into novel post-operative medical devices for treating glioblastoma. This review has the potential to provide a solid grounding for medical researchers and engineers as well as clinicians treating patients with glioblastoma.

 The paper presents a good introduction to the challenges posed by glioblastoma, which is suitable for non-clinicians. Given the current zeitgeist in cancer research towards CNS tumours in general, and glioblastoma specifically, this review is a timely summary.

 Figures are clear, well-presented and provide a helpful representation of the administration and mechanism of action for novel devices. 

 There are five general comments which would benefit from review prior to publication, along with a number of minor comments detailed below. 

 Firstly, throughout the text the authors seem to overemphasise the evidence provided by feasibility, pilot and phase I studies. The following is a non-exhaustive list of examples of such overemphasis:

  • Regarding convection enhanced delivery the authors state this “has been shown to be more effective compared with the conventional diffusion based approaches” (line 119/120). This appears inconsistent with the evidence provided in Table 1 summarising evidence to date, which only includes pilot and stage 1 trials.
  • The authors state "Overall, clinical studies have demonstrated the safety and efficacy of CED in the treatment of GBM” (line 182/183). This does not appear to be sufficiently backed by evidence limited to phase 1 studies. 
  • Regarding ultrasound mediated BBB the authors state “Multiple preclinical studies have established the efficacy of FUS BBB modulation for drug delivery” (line 201). This does not appear sufficiently backed by evidence limited to phase 1/2a studies. The text and Table 2 do not currently provide statistical evidence for this assertion.  
  • The authors state “The first clinical trial investigating BBBO… demonstrate a safe and effective opening of the BBB” (lines 213-15).  It is made clear in Table 2 that this refers to a phase 1 single arm trial of 5 patients, which would be an insufficient sample size to demonstrate efficacy. 
  • Results in sections 2.2, 2.3, 2.4 and 2.5 are presented without comment on the statistical significance.
  • The authors comment on MagForce technologies approval by the Eurpean medicine agency (lines 333-40) without reference. If studies on the efficacy of MagForce are not available in the published literature this would benefit from comment.
  • The authors comment “LITT is an… effective treatment for GBM” (line 728/729). The evidence presented consists of pilot studies, phase 1 trials and retrospective analysis. Little comment is made on the limitations of this evidence base.
  • In contrast to the above, section 3.1.4 in the authors' field of work provides a clear review of available evidence, with comment on statistical significance. The examples provided above would benefit from similar rigour.

 Secondly, in a number of cases the authors' description of trial outcomes appears to under-emphasise complications and significant side-effects. The following is a non-exhaustive list of examples:

  • The authors comment “Overall LITT was generally well tolerated” regarding the 2013 Sloan Phase 1 trial (line 690). This is followed by a description of two patients who developed “serious neurological side effects”. From the information provided in Table 5 (which is not stated in the text) this study involved 10 patients, with three episodes of transient aphasia, three cases of transient hemiparesis, one case of DVT and one of meningitis.
  • The Patel study is described as showing “relatively few complications” (line 700). Contrary to this description, the information provided in Table 5 indicates that of 87 patients, there were 14 instances of neurological deficits, three haemorrhages, five refractory oedemas, two infections and three deaths.
  • The authors comment “these studies have altogether shown that LITT is well tolerated…” (line 728). The authors describe eight studies including pilot studies, phase 1 trials and retrospective analysis; all document significant rates of severe complications. To crudely summarise the presented studies: 255 patients are included, of which the complications include 41 cases of neurological deficit, six intracranial haemorrhages, five deaths, five instances of DVT, 10 infections (including meningitis and ventriculitis), two hydrocephalus, five cases of refractory oedema and four seizures.
  • Regarding HIFU, the authors detail inadvertent ablation of healthy brain parenchyma in one of three patients (line 384/385). The authors comment that “additional safety measures including software modifications… were implemented to improve safety” (lines 387-89). The clinical reader would benefit from further description of the safety measures taken, and whether these have subsequently been investigated in patient trials.

Thirdly, the authors declare a conflict of interest regarding the development of implantable electric field therapy devices. Whilst the review overall presents a balanced and well-rounded description of novel devices, it is notable that relatively minor side-effects and practical limitations of external electric field therapy have been explored in detail, including moderate skin irritation and disruption to daily activities. This is in contrast to the description of other medical devices. 

Fourthly, the authors provide a good summary of barriers to clinical practice for electric field therapy (lines 524-34). However, the clinical readership would benefit from a similar summary for the other medical devices described. Cost/benefit data would also be of interest to clinicians looking to determine the plausibility of translation into clinical practice. If this information is not available this would benefit from comment in the conclusion.

The final general comment is that it is notable that the authors' description of devices for the treatment of GBM is limited to post-operative treatment. As gross total resection has been shown to improve outcomes in GBM. The plethora of surgical devices currently under investigation to improve glioma resection should be commented upon (including, but not limited to, confocal microscopy, Ramann spectroscopy, hyperspectral imaging, and probe-based quantification of PpIX fluorescence); novel devices for phototherapy are also not commented upon. It may be that this is beyond the scope of this review, in which case this would benefit from clarification within the introduction.

 Finally, there are several minor points which may benefit from correction:

  • Fig. 1 includes inconsistent descriptions of clinical trials, which may hinder interpretation - e.g.: “first clinical study", "pilot trial", "pilot clinical trial", “first published pilot trial”, “first in human study”. This would benefit from standardisation.
  • Line 114/115 describes the first conceptualisation of CED by Bobb et al 1994. This should be referenced
  • Line 125 describes CES agents including “chemotherapeutic agents, viral vectors and monoclonal antibody”; for readability this should be changed to “antibodies".
  • Lines 293-304 describe the use of PpIX as a sonosensetiser. It would be helpful to note that this is routinely given to GBM patients for surgical resection. NICE includes use of PpIX for resection in its standards of care.
  • Table 2 includes a column on "main findings," Table 3 “observations”, Table 4/5 “complications”. This would benefit from standardisation.

Author Response

Dear Reviewer 1,

Thank you for reviewing our submitted review article entitled Medical device advances in the treatment of glioblastoma”.

Thank you very much for your comments and suggestions which we have since endeavoured to incorporate into our revised manuscript. 

We agree that we have overlooked overemphasis on the evidence from feasibility and phase I studies, and we are most grateful that you have highlighted this for revision. We have therefore carefully reviewed the manuscript and revised statements of overemphasis in the examples you have cited, and anywhere else they have occurred.

With regard to under-emphasis of complications and side-effects, we have likewise ensured that these have been appropriately highlighted in relevant sections, particularly when reviewing LITT.

We have also endeavoured at your suggestion to look into potential translation barriers for technologies other than EFT, and also reviewed cost-effectiveness where information is available.

Finally, minor points raised for correction have been addressed.

Once again, thank you for your time and for making insightful suggestions that we are sure will improve the quality and completeness of our review.

Sincerely,

Richard Fu, on behalf of all authors.

Reviewer 2 Report

Dear Authors

Your manuscripts represents an interesting and nearly complete review about innovative treatment strategies for glioblastoma.

Besides some minor aspect I wonder that there no information about the progresses in neurosurgery by using fluorescence guided resection e.g. using the fluorescence of 5-ALA induced PPIX. This way improving the resection of more tumor mass. In 2007, 5-aminolevulinic acid (5-ALA) was approved in Europe by the European Medicines Agency (EMA) (brand name: Gliolan®) for "the visualization of malignant tissue during surgery for malignant glioma (WHO III and IV) in adults." Similarly, approval for 5-ALA was granted by the FDA in 2017 as an "optical imaging agent indicated in patients with gliomas (suspected World Health Organization Grades III or IV on preoperative imaging) as an adjunct for the visualization of malignant tissue during surgery" (brand name: Gliolan®).

Furthermore the effort performed by the French-group within the INDYGO-trial (NCT03048240) to perform PDT in the resection cavity is not mentioned. Related to this PDT in the resection cavity trial, studies and trials using interstitial photodynamic therapy as minimally invasive treatments are under way showing highly intersting and promising results. The technical approaches for iPDT are similar to LITT but light application should be performed in a non-thermal manner, thus light dosimetry is a challenge. iPDT is well-known for its highly selective procedure based on the selective accumulation of 5-ALA induced PPIX within the tumor cells. Here 5-ALA induced PPIX is used as photosensitizer. Clinical trials are running (e.g. NCT03897491) and additional information can be derived from publications like Lietke et al. Interstitial Photodynamic Therapy Using 5-ALA for Malignant Glioma Recurrences. Cancers (Basel). 2021 Apr; 13(8): 1767. doi: 10.3390/cancers13081767.

Regarding LITT on GBM first clinical trials were performed in 1994 (Schwarzmaier-group https://journals.lww.com/jcat/Abstract/1994/07000/MRI_Guided_Laser_Induced_Interstitial.2.aspx and following). Based on that investigations finally there were nowadys two commercially available LITT platforms on the market. There are the “Visualase” system approved by the FDA in 2009 after a phase I study in 4 patients with brain metastasis, and the Monteris NeuroBlate Platform, which was first approved by the FDA in 2013 after a first in man trial in 10 patients with unresectable recurrent glioblastomas. (see: J Neurooncol. 2021; 151(3): 429–442. Published online 2021 Feb 21. doi: 10.1007/s11060-020-03652-z Laser interstitial thermotherapy (LITT) for the treatment of tumors of the brain and spine: a brief review. Clark Chen, Ian Lee, Claudio Tatsui, Theresa Elder, and Andrew E. Sloan)

I think these 3 information should be added in specific related paragraphs to complete the review.

Based on that comments figure 1 should be revised and LITT as well as FGR and PDT should be added at correct times.

In addition figure 2 should be revised adding the FDA/CE/EU-approved FGR on the left arm „Therapy Enhancing“ and on the Novel-arm the minmally-invasive approach of interstitial photodynamic therapy should be added as well.

Minor changes:

As there is a new WHO-GBM classification it should mentioned to what classification this review is referred to – old or new?

L290: Mentioning 5-ALA as a sonosensitizer is not correct. The sonosensitizer is PPIX. Thus a correct wording like „5-ALA induced PPIX“ would be better more correct -> throughout the text

Mpa should be changed to MPa

Manuscript as well as tables should be double checked regarding unfortunate hyphenation (e.g. Neuro-Blate instead of Neu-roBlate, ….) and orthography

Author Response

Dear Reviewer 2,

Thank you for reviewing our submitted review article entitled “Medical device advances in the treatment of glioblastoma”.

Thank you very much for your comments and suggestions which we have since endeavoured to incorporate into our revised manuscript. 

We very much appreciate you bringing to our attention PDT technology, and the recent studies in this field. We apologise for embarrassingly overlooking this technology. We have addressed this by creating a whole new section on PDT (and revised our figures accordingly) in the revised article, which we hope you will like. It is a very interesting technology indeed, and we enjoyed writing about it in detail.

Likewise, we have added your suggestions relating to LITT in the relevant section and thank you very much for bringing the suggested papers to our attention.

With regard to your comment on fluorescence guided resection, we fully agree that this represents important surgical technology that has served to improve patient outcomes. However, because it is a surgical adjunct rather than a device that delivers therapy, we have considered this out of the current review's scope. For clarity, we have revised our introduction accordingly to highlight this distinction for readers.

Finally we have addressed your suggestions for minor changes and made corresponding amendments.

Once again, thank you for your time and for making insightful suggestions that we are sure will improve the quality and completeness of our review.

Sincerely,

Richard Fu, on behalf of all authors.

Round 2

Reviewer 1 Report

Overall the authors have addressed my suggestions. Only one minor point:

  • Results in sections 2.2, 2.3, 2.4 and 2.5 are presented without comment on the statistical significance.

-       This has not been fully addressed in version 2. Whilst not in of itself a barrier to publication the reader would benefit from a comment on the statistical significance of results (for example in line 163  the authors comment “use of CED resulted in higher volume of distribution of reovirus” – it is unclear from the text if this met the threshold of statistical significance. 

Author Response

Dear Reviewer 1,

Thank you for taking the time to review our revisions. 

At your suggestion, we have revisited studies presented in 2.2-2.5 and have either included statistics or highlighted where these were lacking, in further revision of the manuscript.

Finally, thank you for all your insightful help and suggestions, which we believe have served to strengthen the quality of this review article.

Sincerely,

Richard Fu on behalf of all authors.

Reviewer 2 Report

Dear Authors

your manuscript showed a lot of improvement especially the review in this content reflects more the actual state-of-the-art.

It should be mentioned that the cited studies were more or less related to the old WHO-GBM classification and that there is new WHO-classification to which new studies have to deal with. e.g that there is probably a change in the patients cohort.

Furthermore the latest iPDT-manuscript published by Cancers shoudl be addressed: Lietke et al. Interstitial Photodynamic Therapy Using 5-ALA for Malignant Glioma Recurrences. Cancers (Basel). 2021 Apr; 13(8): 1767. doi: 10.3390/cancers13081767.

During reading the manuscript again I realized some formatting (e.g. tables should not be cut by page change) as well as orthographic esp. hyphen mistake:

L11-14: formatting

Keywords: PDT should be included

L59-79: Mentioning FGS should be a must as this is state of the art instead of just white light surgery.

L149: 75% out of line

L211: de-livery?

L245: pa-tients

L249: chemo-therapy

L252: per-patient, con-gestion

Table 2: first row - main finding MPa

L294: vices-- such

L298: EFT..

L398: med-ical

L403: in-terference

L428: deficits..

Figure 5: 630-635nm instead of 630nm (only HPD, DHE) or 635nm (PPIX)

L738: ::Visualase

L742: cur-rent and [151,152] . 

L754: pa-tients

L758: sug-gests

L764: defi-cits

L795: cm³

L807: improve-ment

L811: al-so

L812: over-all

L819-822: formatting

L825: pho-to-activation

L827: pres-ence

L882: cm³

L903: J/cm²

L904: J/cm²

Author Response

Dear Reviewer 2,

Thank you for taking the time to review our revisions and we are pleased to learn that you are happy with the improvements.

Following your suggestions, we have made the following further revisions:

  • Included the latest iPDT-manuscript published by Cancers -- Lietke et al. Interstitial Photodynamic Therapy Using 5-ALA for Malignant Glioma Recurrences. Cancers (Basel). 2021 Apr; 13(8): 1767. We have highlighted key aspects of this important study.
  • Highlighted the importance of FGS as a valuable state of the art adjunct and have further included a reference to an important recent review on this topic.
  • In the discussion, we've highlighted the importance of future studies taking into consideration recent revised 2021 WHO classifications when selecting/stratifying patients.
  • Addressed the points you have raised on formatting. 

Finally, thank you for all your insightful help and suggestions, which we believe have served to strengthen the quality of this review article.

Sincerely,

Richard Fu on behalf of all authors.